# Aptamer-Gated Mesoporous Silica Nanoparticles for N Protein Triggered Release of Remdesivir and Treatment of Novel Coronavirus (2019-nCoV)

**DOI:** 10.3390/bios12110950

**Published:** 2022-11-01

**Authors:** Xiaohui Zhang, Xin Zhang, Aoqiong Xu, Mengdi Yu, Yu Xu, Ying Xu, Chao Wang, Gege Yang, Chunxia Song, Xiangwei Wu, Ying Lu

**Affiliations:** 1Department of Applied Chemistry, Anhui Agricultural University, Hefei 230036, China; 2Key Laboratory of Agricultural Sensors, Ministry of Agriculture and Rural Affairs, Anhui Agricultural University, Hefei 230036, China; 3Key Laboratory of Agri-Food Safety of Anhui Province, College of Resources and Environment, Anhui Agricultural University, Hefei 230036, China

**Keywords:** 2019-nCoV, Remdesivir, targeted therapy, mesoporous silica, aptamer

## Abstract

Since the 2019-nCoV outbreak was first reported, hundreds of millions of people all over the world have been infected. There is no doubt that improving the cure rate of 2019-nCoV is one of the most effective means to deal with the current serious epidemic. At present, Remdesivir (RDV) has been clinically proven to be effective in the treatment of SARS-CoV-2. However, the uncertain side effects make it important to reduce the use of drugs while ensuring the self-healing effect. We report an approach here with targeted therapy for the treatment of SARS-CoV-2 and other coronaviruses illness. In this study, mesoporous silica was used as the carrier of RDV, the nucleocapsid protein (N protein) aptamer was hybridized with the complementary chain, and the double-stranded DNA was combined with gold nanoparticles as the gates of mesoporous silica pores. When the RDV-loaded mesoporous silica is incubated with the N protein, aptamer with gold nanoparticles dissociate from the complementary DNA oligonucleotide on the mesoporous silica surface and bind to the N protein. The releasing of RDV was determined by detecting the UV-vis absorption peak of RDV in the solution. These results show that the RDV delivery system designed in this work has potential clinical application for the treatment of 2019-nCoV.

## 1. Introduction

Since the end of 2019, novel coronavirus (2019-nCoV) or the severe acute respiratory syndrome coronavirus 2 (SARS-CoV-2) has affected almost all countries and infected over 614 million people, and caused approximately 6.5 million related deaths [1,2,3]. SARS-CoV-2 is an enveloped, positive-sense, single-stranded RNA β-coronavirus that has been considered as one of the most highly pathogenic viruses [4,5,6,7]. In the past more than 2 years, SARS-CoV-2 has evolved several variants, and some of them have raised regional infections, especially in cities with high population densities [8,9]. The main variants that cause rapid emergence within populations include B.1.1.7 (Alpha) [10], B.1.351 (Beta) [11], B.1.1.28 (Gamma and Zeta) [12], B.1.617.2 (Delta) [13], B.1.1.529 (Omicron) [14], and so on. Today, the main populate variant is Omicron, which is estimated to be 10-fold more infectious than the original virus [15,16]. However, the severity of symptoms and hospitalization rates caused by Omicron are much lower than the original virus [17,18,19]. The accurate medicinal treatment and correct dosage should be given to the hospitalized patients to decrease the side effects caused by drugs.

Remdesivir (RDV) and some other nucleoside analogues exhibit activity against a broad spectrum of viruses, such as Nipah virus, Middle East respiratory syndrome (MERS-CoV), Ebola virus, and severe acute respiratory syndrome (SARS) [20,21,22,23,24]. RDV is an antiviral prodrug of an adenine derivative that can inhibit SARS-CoV-2 in vitro and therefore inhibits viral RNA replication [25,26,27,28,29,30]. In nonhuman primate studies, RDV effectively reduced lung virus levels and lung damage [31]. Although drugs such as RDV were considered as candidate therapeutics for 2019-nCoV treatment, its side effects, such as anemia, at high doses are uncertain, since the research period is too limited [32]. Strategies that might reduce the side effects as well as administer optimal doses to patients should be developed for the purpose of accurate treatment.

In this study, N protein aptamer-gated mesoporous silica nanoparticles (MSNs) for targeted delivery of RDV is proposed. MSNs are considered to have unique advantages of a porous structure and nontoxicity that enable them to load and deliver drugs in vivo [33,34,35,36,37,38,39,40]. N protein, as a major structural protein of coronavirus, plays an important role in packaging the RNA genome into helical ribonucleoproteins, modulating host cell metabolism, and regulating viral RNA synthesis during replication and transcription [41,42]. Nucleic acid aptamer is a single-stranded oligonucleotide that can bind to its target with high affinity and specificity [43,44,45]. The N protein nucleic acid aptamer and gold nanoparticles (Appendix A of SI) employed here can be used as the gating of mesoporous silica to better target the release RDV. N protein aptamer is marked with a sulfhydryl group at one end to combine with gold nanoparticles, and then hybridizes with the -COOH modified complementary strand to form double-stranded DNA (dsDNA) oligonucleotides (the specific DNA sequences are shown in Appendix A of SI). The dsDNA combines with the RDV-loaded and ${NH}_{2}^{-}$ functionalized MSNs by amide reaction, then the MS channel is closed to obtain the RDV-loaded MSNs encapsulated by gold nanoparticles (MSN-Au). When the MSN-Au are incubated with N protein, the gold-nanoparticle-modified aptamer combines with N proteins and disassociates from the surface of the MSNs, the MSN channel is opened, and the RDV loaded within the MSNs is released. This target-triggered RDV release strategy may pave the way toward establishing a novel treatment strategy that could protect patients from the side effects of drugs.

## 2. Materials and Methods

### 2.1. Materials

Cetyltrimethylammonium bromide (CTAB), fluorescent amine, and tetraethyl orthosilicate (TEOS) was purchased from Shanghai Aladdin Biochemical Technology Co., Ltd. (Shanghai, China). Remdesivir (RDV; 99.74%) was acquired from MedChemExpress (Monmouth Junction, NJ, USA). 3-aminopropyltriethoxysilane (APTES) and 1-(3-dimethylaminopropyl)-3-ethylcarbodiimide hydrochloride (EDC) were obtained from Shanghai McLean Biochemical Technology Co., Ltd. (Shanghai, China). N-hydroxysuccinimide (NHS), immunoglobulin G (IgG), hemoglobin (HGB), fibrinogen (FIB), cytochrome c, thrombin, and lysozyme (LZ) were acquired from Beijing Soleibo Technology Co., Ltd. (Beijing, China). Nucleocapsid protein (N protein) was purchased from Beijing Yiqiao Shenzhou Technology Co., Ltd. (Beijing, China). Gold nanoparticles were provided by Nanjing Xianfeng nano material technology Co., Ltd. (Nanjing, China). T4 polynucleotide kinase (T4 PNK) was purchased from Sangon Biotechnology Inc. (Shanghai, China). Bovine serum albumin (BSA) was acquired from Shandong sikejie Biological Co., Ltd. (Shandong, China). Alkaline phosphatase (AKP) was supplied by Shanghai Biyuntian Biological Co., Ltd. (Shanghai, China). N protein aptamers (5′-GCTGG ATGTC GCTTA CGACA ATATT CCTTA GGGGC ACCGC TACAT TGACA CATCC AGC-SH-3′) and complementary DNA (5′-COOH-TTGTA CTGGC TCATA GCTGG ATGTG TCAAT GTAGC GGTGC CCCTA AGGAA TATTG TCGTA AGCG-3′) chain were synthesized and purified by Sangon Biotechnology Inc. All the solutions were prepared with ultrapure water obtained from a Millipore water purification system (>18.2 MΩ·cm). All other chemicals were of analytical grade.

### 2.2. Instrumentation

Transmission electron microscopy (TEM) images were obtained using a Hitachi HT-7700 transmission electron microscope (Tokyo, Japan). Fluorescent absorption spectra were recorded with a G9800A fluorescence spectrophotometer (Beijing, China). Ultraviolet-visible (UV-vis) absorption spectra were recorded with a TU-1901 UV-vis spectrophotometer (Beijing, China). Infrared absorption peaks were obtained with a Fourier transform infrared spectrometer (FTIR Spectrometer; Thermo iS50, Thermo Fisher Scientific, MA, USA). The synthesis of DNA double-strands was realized by polymerase chain reaction (PCR; Thermal Cycler S1000, Bio-Rad, CA, USA). The specific area and mesoporous analysis of mesoporous silica were obtained from specific surface area and pore size analysis BET (BET; ASAP2460, micromeritics, GA, USA).

### 2.3. Binding of Aptamer to Gold Nanoparticles

The synthetic and quantitatively analyzed N protein aptamers were hybridized with complementary DNA oligonucleotides to form double-stranded DNA (dsDNA) chains by mixing N protein aptamers (20 μM) and complementary DNA (20 μM) into 90 μL of Tris-MgCl_2_ (5 mM MgCl_2_, 140 mM NaCl, 20 mM Tris-base, pH 7.4), and 10 μL of annealing buffer (100 mM Tris, 0.5 mM EDTA, 1 mM NaCl, pH 7.4) was added. Next, the resulting solution was heated to 95 °C in the PCR instrument for 10 min and slowly cooled to 20 °C to obtain dsDNA. The synthesis of DNA double strands was further demonstrated by gel electrophoresis (Appendix A). In brief, 20 μL (0.25 mM) of gold nanoparticles with an average diameter of 5 nM were added to the solution of dsDNA. The gold nanoparticles were mixed with SH-terminated dsDNA for 12 h, and dsDNA with gold nanoparticles (Au-DNA) were obtained by Au-S bonding. Finally, the excess dsDNA strands were removed by centrifugation at 3000 r/min for 10 min in an ultrafiltration tube. The resulting Au-DNA were then resuspended in 80 μL PBS buffer (137 mM NaCl, 2.7 mM KCl, 10 mM Na_2_HPO_4_, 1.75 mM KH_2_PO_4_, 50 mM MgCl_2_, pH 7.4).

### 2.4. Binding of Aptamers Modified with Gold Nanoparticles to MSN-NH_2_

Aminated mesoporous silica was prepared according previous reports [35,46]. Briefly, 3 mg of RDV solid powder was dissolved in 5 mL of PBS and acetonitrile (ratio 10:1), and dissolved uniformly by ultrasound. Then, 10 mg aminated mesoporous silica was added to the RDV solution. The RDV was diffused to the pore size of aminated mesoporous silica by stirring slowly with a magnetic stirrer for 24 h. Afterward, 100 μL of the RDV was mixed with the mesoporous silica and Au-DNA solution. Finally, 10 μL EDC (250 mM) and 10 μL NHS (1.25 M) were added to the mixed solution and oscillated slowly at 10 °C (24 h) to obtain MSN-Au.

### 2.5. Characterization of Mesoporous Silica

The specific surface area, pore volume, and pore size distribution were calculated by BET method with a specific surface area and pore size analysis BET instrument. The samples were dispersed in anhydrous ethanol and dropped onto a copper mesh for adsorption. After drying at room temperature, the pore structure and morphology were observed by transmission electron microscopy. The surface-modified functional groups of MSN-NH_2_ were analyzed by FTIR spectroscopy using the KBr tablet method. A standard curve was established between fluorescent amine and APTES, and the amino concentration on mesoporous silica surface was calculated by measuring the sample.

### 2.6. N Protein Aptamer Selectivity

Nine common blood disruptors, i.e., AKP (20 U∙mL^−1^), BSA (2 μM), T4 PNK (20 U∙mL^−1^), IgG (2 μM), thrombin (2 μM), HGB (2 μM), LZ (2 μM), cytochrome C (2 μM), FIB (2 μM), and N protein (2 μM) were selected to evaluate the binding specificity of aptamers to N protein. The release of RDV was measured by detection of the peak UV-vis signal at 247 nm.

### 2.7. Analysis of RDV-Loaded Mesoporous Silica in Real Samples

Five volunteers were randomly selected. The selected volunteers included two male members and three female members, aged between 20 and 50 years. Blood samples were obtained by drawing venous blood from five volunteers and placing the blood into an anticoagulant tube EDTA-K2. Subsequently, 2 μL of blood sample was added into 100 μL of MSN-Au solution. Then, different concentrations of N protein were added to the mixed solution of blood and MSN-Au, and the RDV release of MSN-Au was detected by UV-vis.

In the above experiment, MSN-Au solution needs to be centrifuged through ultrafiltration tube and cleaned with PBS to remove excess RDV.

## 3. Results and Discussion

The mechanism of target-triggered RDV releasing from MSN-Au is depicted in Figure 1. MSNs were first synthesized by CTAB, and the RDV molecules were loaded into the pores by oscillating on a shaker with magnetic stirrer. When RDV-loaded mesoporous silica binds to gold nanoparticles modified double-stranded DNA, the pores of MSNs are enclosed by gold nanoparticles. After being challenged with N protein, the aptamer linked with gold nanoparticles dissociates from its complementary DNA oligonucleotide and combines with the target. As the gates of MSNs are opened, the RDV is released from the pores and the targeted therapy is expected to be realized.

### 3.1. Mesoporous Silica Characterization

The synthesized MSNs were characterized by TEM and SEM imaging, the diameter of typical MSNs is in the range of 50 to 100 nm (Figure 1A,B and Appendix A). As shown in Figure 1B, there are large number of even pores arranged on the MSNs, which provides sufficient space for RDV loading. The surface area of the MSNs was tested by nitrogen adsorption–desorption isotherms and calculated to be 760.6 m^2^·g^−1^ (Figure 1C), and the average pore size was 2.7 nm (Figure 1D). The three-dimensional size of RDV was calculated to be 1.8, 1.2, and 1.1 nm, respectively (Appendix A). These results indicate that the pore size of MSNs is sufficient for RDV loading. The FT-IR spectrum of MS-NH_2_ is shown in Figure 1E. The typical O-Si-O bending vibration peak can be seen at 467 cm^−1^, and the peak at 1091 cm^−1^ was caused by the stretching vibration of Si-O-Si. The bending vibrations of Si-OH in aminoated mesoporous silica correspond to peaks at 961 and 1641 cm^−1^. The vibration peak of -NH- was at 1503 cm^−1^. The absorption peak at 3427 cm^−1^ was caused by O-H stretching vibrations of water molecules. The above absorption peaks at 3427, 1641, 1503, 1091, 961, and 467 cm^−1^ match well with the characteristic absorption positions, which confirms the successful synthesis of aminoated mesoporous silica. The concentration of amino on mesoporous silica was 0.63 mM by fluorescence detection (Figure 1F). Changes in X-ray diffraction XRD peaks before and after amino-modified mesoporous silica are presented in Appendix A.

### 3.2. UV-vis Absorption Linear Range and Detection Limit of RDV

The concentration of RDV in PBS solution could be determined by UV-vis method. As shown in Figure 2A, the baseline represents the UV-vis diagram without RDV, and the concentrations of RDV added from bottom to top are 1 mg·L^−1^, 3 mg·L^−1^, 5 mg·L^−1^, 7 mg·L^−1^, 10 mg·L^−1^, 12 mg·L^−1^, 15 mg·L^−1^, 17 mg·L^−1^, and 20 mg·L^−1^, respectively. The absorption signal intensity at 247 nm increased with the increasing concentration of the RDV. The relationship between UV-vis intensity and RDV concentration was linear in the range of 1–20 mg·L^−1^, and the linear regression equation was A = 0.055 + 0.036C_RDV_, R^2^ = 0.995 (Figure 2B). The detection limit was 0.5 mg·L^−1^ (S/N = 3).

### 3.3. RDV Loading of Mesoporous Silica

To investigate the target-triggered RDV release behavior of MSN-Au, UV-vis and TEM were conducted. As shown in Figure 3A, with UV-vis detection, the blank PBS solution without MSN-Au shows no peak signal (grey line). The MSN-Au solution shows an absorption peak at 247 nm (green line), while the peak signal obviously increased upon N protein addition (blue line). After adding 2 μM N protein, the absorption peak at 247 nm was 4.9 times stronger than the background signal. The changes in MSN-Au before and after incubation with N protein were further studied by TEM. In the case of gold nanoparticle-capped MSNs, dark spots on the shells of the MSNs were observed in the TEM microscopy (Figure 3B). After incubation with N protein, the dark spots on the surfaces of the MSNs decreased significantly (Figure 3C). These results indicate that the gold nanoparticle-capped gates on the MSNs could be opened after the N proteins were added, and the RDV release progress was realized. The MSN-Au stability was tested and the result is shown in Appendix A.

### 3.4. Aptamer Selectivity

In order to verify the unique selectivity of aptamer for N protein, nine proteins commonly found in blood (AKP, BSA, T4 PNK, IgG, thrombin, HGB, LZ, cytochrome c, FIB) were selected as interfering substances. The individual interfering protein and all the interfering mixtures (Appendix A of SI) were then reacted with MSN-Au, respectively, to evaluate the effect of aptamer on the selective delivery process. Figure 4 shows the UV-vis absorption peak at 247 nm after the MSN-Au solutions were incubated with AKP (20 U∙mL^−1^), BSA (2 μM), T4 PNK (20 U∙mL^−1^), IgG (2 μM), thrombin (2 μM), HGB (2 μM), LZ (2 μM), cytochrome C (2 μM), FIB (2 μM), and N protein (2 μM). Compared with 2 μM N protein, the absorption peak signal obtained on the UV-vis when challenged with the nine interferences were 11.62%, 9.96%, 14.94%, 7.05%, 6.64%, 6.22%, 12.86%, 9.13%, and 4.56%, respectively.

### 3.5. Application Analysis in Human Blood

This work attempted to detect RDV release of MSN-Au in human blood. Five volunteers were randomly selected, and their clinical characteristics are shown in Appendix A. Blood was not pretreated before adding MSN-Au. We added 0 μM, 0.5 μM, 1 μM, and 2 μM of N protein to the mixed solution of blood and MSN-Au, respectively. From Figure 5, it can be seen that the UV-vis absorption peak gradually increases with the increasing of N protein concentration in the mixed solution. The correlation between concentration of RDV releasing and the amount of N protein addition is shown in Appendix A. Statistical methods were used to obtain the average recovery of N protein in blood samples. Recovery experiments were performed by adding N protein at 0.5 μM, 1 μM, and 2 μM (as shown in Appendix A).

## 4. Conclusions

We constructed a gated drug delivery system based on mesoporous silica as a drug carrier. In this work, aptamers and gold nanoparticles were employed as gates of mesoporous silica, which is expected to reduce the side effects of RDV. Compared with the nine interfering proteins, the aptamer of N protein showed stronger selectivity to N protein. The drug delivery vector designed in this study achieved good results in the blood samples, with different concentrations of N protein added to the samples of five volunteers. This target-induced treatment system is expected to be applied in clinical therapy of 2019-nCoV.

## Data Availability

Not applicable.

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
