# Peer review of "Aptamer-Gated Mesoporous Silica Nanoparticles for N Protein Triggered Release of Remdesivir and Treatment of Novel Coronavirus (2019-nCoV)"

_biosensors, 2022, doi:10.3390/bios12110950_

Round 1

Reviewer 1 Report

Zhang et al. demonstrated the detection of Corona virus by using aptamer gated mesoporous silica nanoparticles. Authors have used targeted therapy for the treatment of SARS-CoV-2. In this study, mesoporous silica was used as the carrier of the Remdesivir, the nucleocapsid protein (N protein) aptamer was hybridized with the complementary chain, and the double-stranded DNA was combined with gold nanoparticles as the gates of mesoporous silica.

Few points need to be considered

1.     What is the specificity of developed sensor in terms of Corona virus with other disease-causing viruses.

2.     Provide aptamer sequence

3.     What is the reason of using both gold and silica nanoparticles

4.     Author should show the characterization of gold nanoparticles also

5.     Please add statistical analysis to graph   

Author Response

Please see the attachment."

Reviewer 2 Report

In this manuscript, the authors developed a trigger release of Remdesivir for the treatment of 2019-nCoV using mesoporous silica as the carrier, the nucleocapsid protein (N protein) aptamer as a targeting for 2019-nCoV. The binding between N protein aptamer with N protein will remove the gold nanoparticles and thus induce the release of Remdesivir. This manuscript needs to perform more experiments to be published. Here are the details comments.

1. Please provide the fold enhancement data after addition of N protein in Figure 3A.

2. It is hard to be convinced that the gold nanoparticles capped gates could be opened after the N protein addition simply by comparing the brightness of the dark spots on the surface of MSNs. Please utilized other more convincing method such as fluorescence labeling of gold nanoparticles etc.

3. Please provide all the interfering proteins data in Figure 4.

4. Please provide the kinetics data for releasing of Remdesivir. Especially they are using fully complementary DNA sequences. Have they tried partially complementary sequences to achieve a faster release?

5. Have they optimized the loading capacity of gold nanoparticles by varying the pore size etc?

6. What is the concentration used for Remdesivir in clinics? Is the released amount fulfilling the requirement?

7. Please provide the cytotoxicity data for the MSNs since they want to treat Covid in the future.

8. There are many grammar and typo issues.

Round 2

Reviewer 1 Report

Satisfactory revision

Reviewer 2 Report

The manuscript can be accepted now.